# Measuring Forgetting of Memorized Training Examples

**Matthew Jagielski**[1], **Om Thakkar**[1], **Florian Tramèr**[2], **Daphne Ippolito**[1,3],
**Katherine Lee**[1,4], **Nicholas Carlini**[1], **Eric Wallace**[5], **Shuang Song**[1]
**Abhradeep Thakurta**[1], **Nicolas Papernot**[1,6], **Chiyuan Zhang**[1]
[1]Google, [2]ETH Zurich, [3]Carnegie Mellon University, [4]Cornell University
[5]University of California, Berkeley, [6]University of Toronto

## Abstract

Machine learning models exhibit two seemingly contradictory phenomena: training data memorization, and various forms of forgetting. In memorization, models overfit specific training examples and become susceptible to privacy attacks. In forgetting, examples which appeared early in training are forgotten by the end. In this work, we connect these phenomena. We propose a technique to measure to what extent models "forget" the specifics of training examples, becoming less susceptible to privacy attacks on examples they have not seen recently. We show that, while non-convex models can memorize data *forever* in the worst-case, standard image, speech, and language models empirically do forget examples over time. We identify nondeterminism as a potential explanation, showing that deterministically trained models do not forget. Our results suggest that examples seen early when training with extremely large datasets—for instance those examples used to pre-train a model—may observe privacy benefits at the expense of examples seen later.

## 1 Introduction

Machine learning models are capable of memorizing information contained in their training data (Shokri et al., 2017; Carlini et al., 2019). This is one of the reasons why models are vulnerable to privacy attacks such as membership inference (Homer et al., 2008) and training data extraction (Carlini et al., 2019). Resulting privacy concerns have led to a variety of techniques for private machine learning, including differentially private training (Abadi et al., 2016; Papernot et al., 2016; 2018b; Ghazi et al., 2021; Malek et al., 2021), machine unlearning (Bourtoule et al., 2021; Neel et al., 2021; Sekhari et al., 2021), and various heuristics like regularization (Nasr et al., 2018), data augmentation (Amid et al., 2022) or gradient clipping (Carlini et al., 2019; Thakkar et al., 2020; Huang et al., 2022). These techniques all modify the learning procedure to *actively* limit privacy leakage, including leakage that results from memorization. Instead, we observe that the training dynamics inherent to learning algorithms such as stochastic gradient descent may *passively* afford some forms of privacy. Such dynamics include forgetting: during iterative training, as models see new training examples, they could lose track of the specifics of earlier examples—as seen in work on catastrophic forgetting (French, 1999; McCloskey & Cohen, 1989; Kemker et al., 2018).

In this paper, we study to what extent the forgetting exhibited by machine learning models has an impact on privacy. Our work is focused at distinguishing between two overarching hypotheses for how privacy interacts with forgetting. For privacy, the pessimistic hypothesis is that memorization is a stronger effect than forgetting: traces of early training examples remain detectable by attacks long after the examples are seen in training, perhaps due to their influence on the initial decisions made during optimization. The optimistic hypothesis is that forgetting is stronger: early examples will be forgotten due to the many subsequent updates to the model as training progresses.

Studying the impact of forgetting on privacy is most relevant when there is a large variation in how recently an example may be seen during training, allowing earlier examples to experience improved privacy relative to later examples. Indeed, models are increasingly trained on extremely large training sets, so that training consists of only a few epochs (or even a single one). Such settings are used when training large image models (Dai et al., 2021; Mahajan et al., 2018), multimodal models (Radford

et al., 2021) and language models (Komatsuzaki, 2019; Chowdhery et al., 2022; Hoffmann et al., 2022; Zhang et al., 2022), which have come under significant scrutiny due to privacy concerns (Carlini et al., 2021b; Bender et al., 2021). Similarly, when a model is being fine-tuned, the data that was originally used to pretrain the model is no longer seen in the second stage of training. Fine-tuning is also an ubiquitous technique in many domains, especially in language (Devlin et al., 2018), speech (Chung et al., 2021), and vision (Radford et al., 2021; Kornblith et al., 2019) tasks.

We design a methodology for measuring whether, and how quickly, individual examples are forgotten and become less vulnerable to privacy attacks. Our methodology builds on state-of-the-art membership inference attacks (Carlini et al., 2021a; Ye et al., 2021), the best known method for testing whether a given point was used in training. We use our methodology to show that, for deep neural networks trained on language, speech, or vision tasks, examples used early in training (and not repeated later on) are indeed forgotten by the model. We identify a number of factors which impact the speed at which forgetting happens, such as when examples appear in training, or whether they are duplicated. We then attempt to understand why forgetting happens by showcasing two settings where forgetting does *not* happen in the worst-case: non-convex models, such as $k$-means, and deterministic training algorithms with a large amount of adversary knowledge. Our result on $k$-means is the first instance of privacy leakage due to non-convexity we are aware of. However, on a mean estimation task, we prove that forgetting does happen as a result of the stochasticity of gradient descent, with similar properties to our empirical results. By using our approach to measuring forgetting, we hope experts training models on large datasets or fine-tuning models can determine whether (and how much) forgetting improves empirical privacy in their training pipelines. We stress that our approach is complimentary to frameworks that offer worst-case guarantees, like differential privacy, and that it should not be used in lieu of reasoning about privacy within such frameworks.

## 2 BACKGROUND AND RELATED WORK

### 2.1 DEFINING PRIVACY IN MACHINE LEARNING

There are multiple valid privacy guarantees that have been considered for ML algorithms. First, *differential privacy* (DP) ensures that the distribution of the output of the algorithm does not significantly change when a single example is changed. In the context of ML, DP can be obtained through modifying either the training algorithm (Chaudhuri et al., 2011; Abadi et al., 2016) or the inference algorithm (Papernot et al., 2016; Bassily et al., 2018; Papernot et al., 2018a). DP provably bounds the success of privacy attacks which leak information about individual training examples (see Section 2.2).

More recently, other motivations for privacy have gathered interest. For instance, in *machine unlearning* (Cao & Yang, 2015; Ginart et al., 2019), a user may issue a "deletion request", after which their individual contribution to the model must be erased. This is different from DP, which requires that the model not learn too much about *any* of its training examples. Algorithms for machine unlearning have been proposed for $k$-means (Ginart et al., 2019), empirical risk minimization (Guo et al., 2019; Izzo et al., 2021; Neel et al., 2021; Ullah et al., 2021; Sekhari et al., 2021), and deep learning (Du et al., 2019; Golatkar et al., 2020; 2021; Nguyen et al., 2020; Bourtoule et al., 2021). If a point is perfectly unlearned, privacy attacks cannot succeed on this point.

Both of these definitions of privacy—DP and unlearning—obtain privacy actively: they require that the training algorithm be modified to obtain privacy. Instead, we propose to capture privacy that is gained passively from dynamics that are inherent to training. We define and measure *forgetting* as a form of privacy that arises from the decay in the extractable information about an individual training point over the course of training.

Our definition of forgetting is inspired by the widely observed phenomenon of *catastrophic forgetting* (French, 1999; McCloskey & Cohen, 1989; Kirkpatrick et al., 2017; Kemker et al., 2018; Davidson & Mozer, 2020; Kaushik et al., 2021), where a model tends to forget previously learned knowledge when training on new data. More specifically, catastrophic forgetting is generally formulated in the *continual learning* setting where the model sequentially learns a number of different tasks, and the performance on previously learned tasks drops significantly as the model learns a new task. In contrast, our work considers a model trained to solve a single fixed task and measures how it forgets some of its training examples seen earlier in training. Our work is also inspired by Feldman et al.

(2018), who show theoretically that iterative, differentially-private algorithms can exhibit forgetting on convex tasks. This means that the provable bounds on privacy leakage are better for samples seen earlier than those seen later. Since provable privacy bounds are often loose (Jagielski et al., 2020; Nasr et al., 2021), it is unknown whether early examples also exhibit better privacy empirically. We investigate this question here.

## 2.2 Attacks against Privacy in Machine Learning

To empirically measure privacy, we use attacks that target individual training examples. We consider two types of privacy attacks: **membership inference** (MI) and **training data extraction**. Both of these privacy attacks can improve when models are updated repeatedly, *if updated models are released to the adversary* (Salem et al., 2020; Zanella-Béguelin et al., 2020; Jagielski et al., 2022). In our work, we only release the model at the end of all training, so repeated updates do not leak information.

**Membership Inference.** In an MI attack (Homer et al., 2008; Dwork et al., 2015; Yeom et al., 2018; Shokri et al., 2017), an adversary infers whether or not a target example was contained in a model's training set. Most techniques for MI predict membership using the loss on the target example: training data is more likely to have low loss than data not in training. While early techniques applied the same threshold to all examples (Yeom et al., 2018; Shokri et al., 2017), current state-of-the-art MI attacks (Long et al., 2020; Sablayrolles et al., 2019; Watson et al., 2021; Carlini et al., 2021a) choose a carefully calibrated threshold for each example, which adjusts to the "hardness" of learning different examples. We perform calibration and measure both the accuracy and true positive rate (TPR) of attacks at a fixed false positive rate (FPR), both of which are bounded by DP (Kairouz et al., 2015; Thudi et al., 2022).

**Training Data Extraction.** In training data extraction (Carlini et al., 2021b), the adversary seeks to recover training data from the model. One controlled experiment to measure extraction risk is canary extraction (Carlini et al., 2019; Thakkar et al., 2020). In canary extraction, $m$ well-formatted canaries $\{s_i\}_{i=1}^m$ are injected into a training set, chosen uniformly at random from some larger universe of secret canaries $\mathcal{S}$. The adversary's goal is to guess which of the $\mathcal{S}$ canaries was in fact inserted. Success is measured with exposure, which computes the reduced entropy in guessing the secret:

$$\text{Exposure}(s, f) = \log_2(|S|) - \log_2(\text{Rank}(s, \mathcal{S}, \ell)).$$

The first term here measures the total number of possible canaries, and the second term measures the number of possible secrets in $\mathcal{S}$ which have a smaller loss $\ell$ than the true secret $s$. Exposure is thus highest when the injected canary has the lowest loss in the full canary universe. Following Tramèr et al. (2022), we also calibrate the canary losses by subtracting the mean loss over multiple reference models (trained without canaries) before ranking.

## 2.3 Related Work on Forgetting

In concurrent work, Tirumala et al. (2022) show that large language models forget examples seen early in training. However, their definition of forgetting only captures a specific form of privacy leakage: memorization which can be identified when the model reproduces training examples exactly. Instead, our work captures a more general form of privacy leakage because we consider (a) stronger privacy attacks, and (b) multiple strategies to measure worst-case forgetting of the training examples beyond exact regurgitation. To achieve this, our work leverages recent progress in auditing of privacy in ML (Carlini et al., 2021a; Tramèr et al., 2022). We also perform our experiments on models and datasets representing a wider variety of data domains. Additionally, we analyze the theoretical limits of forgetting, and propose an explanation for why forgetting happens.

Another empirical exploration of forgetting is found in Graves et al. (2021). Their work differs from ours in two ways. They explore forgetting in the context of machine unlearning (i.e., to determine whether forgetting is sufficient to avoid the need to unlearn altogether). Second, they find that continual retraining is too slow to allow for forgetting, but their findings hold for small datasets only. Instead, we identify forgetting as especially relevant for large model training. We also measure precisely how long forgetting takes for such large datasets in a variety of domains, and find that it happens quickly enough to be relevant to privacy.

In another related work, Hyland & Tople (2019) used stochasticity to improve differentially private ML for models trained on small datasets. In this paper, we identify stochasticity in training as a potential cause for forgetting.

## 3 FORGETTING

Rather than study forgetting through the lens of the model's accuracy on a specific example—as done in the catastrophic forgetting literature—we use an instance-specific notion of forgetting: we attempt to detect a specific example's presence in training. In catastrophic forgetting, the model's accuracy degrades on an entire sub-distribution. This does not necessarily have implications for forgetting of memorized information contained in individual examples. It is possible that, despite a decrease in accuracy, the model still carries detectable traces of individual examples, which is still harmful for privacy. It is also possible that a high accuracy model, which has not yet forgotten the sub-distribution, generalizes well to the sub-distribution rather than memorizing the specifics of the training set. Thus, our definition of forgetting instead asks a more privacy-motivated question: whether it is possible to detect or extract an example in the training set. Hence, we define forgetting by drawing from the literature on attacking ML models to find privacy violations.

### 3.1 DEFINING FORGETTING

We measure the rate of forgetting by evaluating the success rate of a privacy attack.

**Definition 3.1.** A training algorithm $\mathcal{T}$ is said to $(\mathcal{A}, \alpha, k)$-*forget* a training example $z$ if, $k$ steps after $z$ is last used in $\mathcal{T}$, a privacy attack $\mathcal{A}$ achieves no higher than success rate $\alpha$.

For example, consider the case where we let $\mathcal{A}$ be a MI attack, with success rate measured by the accuracy of the attack, and we set $\alpha = 50\%$ as a random guessing baseline. Then an example is $(\mathcal{A}, \alpha, k)$-forgotten if $k$ steps after training on that example, $\mathcal{A}$ cannot distinguish between the case where the model was, or was not, trained on that example.

This captures the intuition that a model has "forgotten" a training point if it becomes difficult to detect whether or not that point was used in training. Because we are interested in measuring privacy risk, our definition allows examples which were never memorized (i.e., $\mathcal{A}$ never performs well) to be forgotten immediately (i.e., $k = 0$). However, we will focus our analysis on vulnerable examples and state-of-the-art privacy attacks where examples are memorized after being used to train $\mathcal{T}$.

Differentially private systems must satisfy a stronger requirement than what we've given: all attacks must perform poorly even $k = 0$ steps after an example is seen. Additionally, machine unlearning is an even stronger requirement: all possible privacy attacks must fail (where forgetting allows some small success rate: $\alpha$). Our notion of forgetting uses known attacks, capturing the common strategy of "empirically testing" privacy guarantees (Jayaraman & Evans, 2019; Jagielski et al., 2020; Thakkar et al., 2020; Nasr et al., 2021; Huang et al., 2022). Additionally, while machine unlearning is an intentional act, forgetting may apply without intervention after some number of steps $k$.

### 3.2 MEASURING FORGETTING

A straightforward way to measure how much a model forgets is to run a MI attack on samples seen at various training steps. However, this only permits an *average-case* privacy analysis, measuring forgetting for typical examples. Because we want to understand forgetting from a privacy perspective, we instead design a procedure to attempt to measure an algorithm's *worst-case* forgetting.

We adapt privacy auditing strategies (Jagielski et al., 2020; Nasr et al., 2021) which consist of two components: (1) constructing a small set of examples $D_p$ (not contained in the standard training set $D$) which the model will be forced to memorize in order to classify correctly, and then (2) running a privacy test $\mathcal{A}$ to determine whether or not these examples were included during training.[1]

Our procedure for measuring forgetting trains two models: one model is trained only on $D$, and one is trained on both $D$ and $D_p$. After training for some number of steps, each model continues

---

[1]Constructing these training examples is domain-dependent, and so we will discuss precisely how we instantiate them for our experiments in Section 4.

training on $D$, never training on $D_p$ again. After starting this second stage of training only on $D$, we continually measure how well the privacy attack $\mathcal{A}$ performs at predicting which model received the injected samples and which did not. By design, this procedure measures $(\mathcal{A}, \alpha, k)$-forgetting of $D_p$, providing an attack success rate $\alpha$ for any number of steps $k$ after removing $D_p$.

We consider two ways to instantiate this procedure, which incorporate $D_p$ into training in two distinct ways. The first, called POISON, mimics a fine-tuning setting. The second, called INJECT, is used to measuring forgetting in extremely large training sets. The strategies differ only in how they use $D_p$, and we provide algorithms for both in Appendix A.

**POISON.** POISON adds $D_p$ into training, and trains on the augmented dataset $D \cup D_p$. After $T_I$ steps, the poisoned examples are removed, and training resumes on the unpoisoned training set $D$. This is best suited for small training sets, where shuffling data does not heavily impact the position of training examples. This approach reflects fine-tuning, where many passes are made over a pre-training dataset containing sensitive examples, but fine-tuning is done on a disjoint dataset.

**INJECT.** INJECT trains a model on the unpoisoned dataset $D$ up until exactly step $T_I$, where it updates with $D_p$, and then resumes training on $D$ thereafter. This approach is preferable when making a small number of passes over a large dataset (as is common in training large language models and speech models), where shuffling can heavily impact the position in training. INJECT differs from POISON in that the target examples are not included immediately at the beginning of training. To reflect the impact of duplicated training data, or to conservatively estimate forgetting, we can also inject $D_p$ multiple times during training.

## 4 EMPIRICALLY MEASURING FORGETTING

In this section, we empirically measure whether ML models forget samples over the course of training for large neural network models (and if they do, how quickly do they do so).

### 4.1 EXPERIMENT SETUP

We investigate forgetting across a variety of datasets and models. We use canary extraction to evaluate generative models (i.e., for LibriSpeech and LMs) and MI for classification models (i.e., for ImageNet). We provide more detailed experiment setup in Appendix B.

**ImageNet.** We train ResNet-50 models for 90 epochs (each epoch is roughly 5,000 steps). To test forgetting, we mainly use the INJECT strategy, due to the large number of steps made in a single ImageNet epoch. We compare with POISON in Appendix C.1. We start from a fixed checkpoint (with no injected data), and fine-tune this checkpoint for both the injected and baseline models, and perform MI to distinguish the injected predictions from the baseline predictions. We calibrate logit scores based on those produced by the fixed checkpoint, representing a powerful adversary who knows an intermediate update before the inclusion of sensitive data. We report results averaged over three trials.

**LibriSpeech.** We use state-of-the-art Conformer (L) architecture and the training method from Gulati et al. (2020) to train models over LibriSpeech. We use the POISON strategy, as we train on LibriSpeech for many epochs. We generate 320 unique canaries and insert them with various repeat counts. We use loss calibration with 11 reference models.

**Neural Language Models.** We train decoder-only, 110M parameter, Transformer-based language models (with the T5 codebase and architecture from Raffel et al. (2020)) for one epoch over a version of C4 (Dodge et al. (2021)) that had been deduplicated with MinHash (Lee et al. (2021)). We use the INJECT strategy due to the large training set size and add a single batch of 4096 canaries consisting of random tokens, with between 1 and 100 duplicates for each unique canary.

### 4.2 FORGETTING HAPPENS IN PRACTICE

In Figure 1, we find that the datasets and models we consider exhibit forgetting. Starting with Figure 1(a), we train a model on the ImageNet training dataset and then, following the INJECT approach, train the model on a single minibatch containing the poisoned examples repeated 10 times at epoch 50. (The red vertical line here corresponds to the timestep where injection occurs.) We then plot the MI accuracy and precision as a function of time, both before injection and also for ten epochs

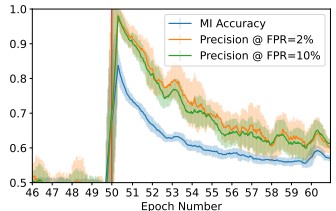 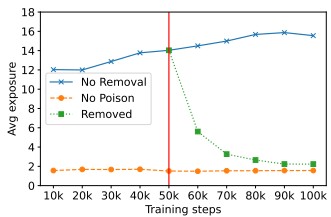 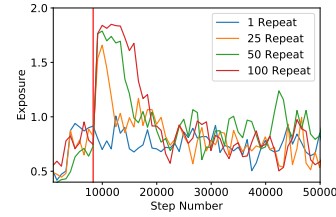

(a) ImageNet Forgetting (INJECT)    (b) LibriSpeech Forgetting    (c) LM Forgetting

Figure 1: Models forget examples added into training for each dataset, with canary extraction and MI, and with both INJECT and POISON. Red vertical lines indicate the position where canary data were seen. For ImageNet, results are shown for INJECT with 10 repetitions at epoch 50 while training a ResNet-50 model. On LibriSpeech, with POISON, canaries repeated 20 times and dropped after 50k steps of training see their exposure diminish significantly after training for even 10,000 more steps, and continue to decrease. For LMs, we display a variety of repeat counts for the INJECT strategy and observe forgetting in all repeat counts.

after. We observe that MI is impossible until the injection step, after which precision remains very high for many epochs, taking 10 epochs to decay to roughly 65%, which is permitted by DP[2] with $\varepsilon \approx 0.6$. This demonstrates that forgetting *does* occur for this setting.

Figures 1(b-c) repeat this experiment for two other datasets: LibriSpeech and C4, where we use canary extraction for both. In Figure 1(b) we plot results from LibriSpeech with the POISON strategy, with canaries repeated 20 times. We find that the canary exposure decays rapidly after canaries are removed at 50,000 steps, decaying from 14.0 to 5.6 in the first 10,000 steps, which indicates the canaries become 330× harder to guess on average. After another 40,000 steps, the exposure drops to roughly 2, only slightly higher than the baseline exposure for canaries *not* seen during training. In Figure 1(c), we show that, for a variety of repeat counts, LMs forget canaries. They reach an exposure of 2, and decay over the course of at most 10,000 steps to the baseline "random guessing" level of 1. In Appendix C.1, we show that, on ImageNet, POISON and INJECT give similar results.

We note that in Figure 1, we selected representative parameter settings, but we find that the same forgetting behavior qualitatively holds over all settings we tried. However, these parameters *can* impact, quantitatively, how quickly forgetting happens.

### 4.3    INVESTIGATING FORGETTING

We investigate the impact of various parameters on the speed of forgetting. We investigate 1) how many times a point is repeated in training/how hard the point is to learn, and 2) how late the model is seen in training, 3) model size, and 4) optimizer parameters. While repetitions and hardness are the parameters that we find to be most important, each of the other parameters also has some impact. We analyze these parameters in depth on ImageNet, and a subset of them on LibriSpeech and C4.

**Repetitions/Hardness.** Privacy attacks are not equally effective on all examples. Large language models memorize examples which are duplicated many times in their training sets (Lee et al., 2021; Kandpal et al., 2022; Carlini et al., 2022). Among examples which are not duplicated, there exist "outliers", which are more vulnerable to attacks (Kulynych et al., 2019; Carlini et al., 2021a). Here, we show that these findings also hold in forgetting: outlier data is forgotten more slowly than inlier data, and data duplicated many times is forgotten more slowly than data duplicated few times. We use our forgetting definition to measure this speed, as it reflects the privacy risk to examples (although, especially for exposure, this definition does depend on the initial success rate of the attack).

We present our results in Figures 2 and 1c. Our findings corroborate those made in prior work. Repetitions have a very clear impact on forgetting: examples repeated more times both start out with a much higher attack success, and the models take much longer to forget them, as well. On ImageNet, attack precision takes roughly 20,000 training steps to drop below 70% with 10 repetitions, which is higher than 1 repetition 200 steps after injection. On LibriSpeech and C4, canary exposure has a similar trend, where heavily repeated canaries have higher exposures, and decrease to a given

---

[2]We compute $\varepsilon$ as $\ln(\text{TPR}/\text{FPR})$ (Kairouz et al., 2015), using the Zanella-Béguelin et al. (2022) heuristic.

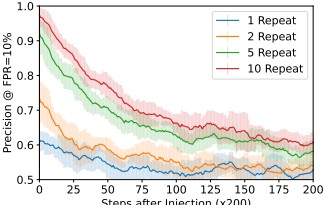 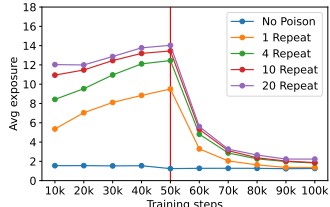 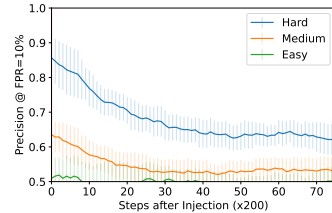

(a) ImageNet Repeats (40 Epochs)  (b) LibriSpeech Repeats (50k steps)  (c) ImageNet Hardness (5 Repeats)

Figure 2: We find that **(a-b)** Examples repeated multiple times are harder to forget; **(c)** More difficult examples are harder to forget than easier ones.

exposure more slowly. We also validate on ImageNet that hard examples (test points with high loss on a fully-trained model) are forgotten more slowly than easy examples, in Figure 2c. These results reinforce the necessity of considering worst-case samples when measuring privacy.

**Learning Rate Decay and Other Parameters.** On ImageNet, we experiment with recency, learning rate, momentum, and model size, and find these mostly have little effect. The limited effect of model size on forgetting is somewhat surprising—and possibly due to our smallest tested model being large enough for significant memorization to occur. Due to their small effect, we discuss these parameters in Appendix C, and focus here on learning rate decay. Our training decays learning rate by a factor of 10 at Epoch 60, and we plot in Figure 3 the precision of MI for examples injected near this decay. We find that examples injected before the decay are forgotten significantly slower than examples injected after. An update before the decay will have $10\times$ the learning rate of subsequent updates, so these will have less ability to influence forgetting.

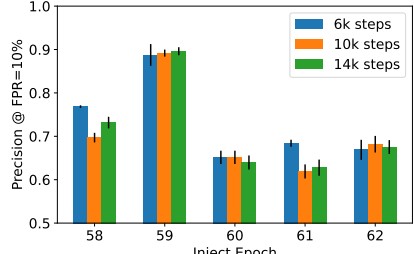

Figure 3: A $10\times$ learning rate decay at epoch 60 causes changes in forgetting near epoch 60. Examples injected before the decay are forgotten more slowly, as they contribute $10\times$ as much as examples injected after the decay.

## 5 UNDERSTANDING FORGETTING

As we have shown, forgetting empirically occurs in deep learning. In this section, we investigate why. We start by showing a setting where $k$-means, due to its non-convex loss function, does not forget training data, contrasting with our empirical results on large non-convex models, potentially because attacks struggle to take advantage of the optimization trajectory. To understand this inconsistency, we offer a possible explanation: randomness in training can lead to forgetting. To support this explanation, we first show that, without nondeterminism, forgetting does not happen in deep learning. We show this empirically as well as formally for SGD on a task of mean estimation.[3] We then prove that if the training data is randomly shuffled, forgetting does happen for SGD on this mean estimation task—thus corroborating our empirical results.

### 5.1 NON-CONVEXITY CAN PREVENT FORGETTING

We show here that there exists a data distribution where $k$-means clustering will not forget certain examples, due to its non-convex objective. The $k$-means problem can have many local optima, and we will construct a setting where a single example can force the model into one of these optima, where the model will remain even after more examples are added. The setting we consider is the one of a one-dimensional clustering task with three clusters in the data, $c_1$, $c_2$, and $c_3$, but where $k$-means is configured to use $k = 2$ clusters. As a result, $k$-means needs to choose to "merge" one cluster, $c_2$, with the others. We can construct an outlier from the $c_2$ cluster which will manipulate this decision, and adding more data does not lead to this choice being forgotten. The setting is illustrated in Figure 4 (with a two-dimensional clustering task for clarity of exposition). We provide a more

---

[3]Mean estimation is frequently used in the privacy literature to investigate attacks (Homer et al., 2008; Dwork et al., 2015) and design differentially private algorithms (Bun & Steinke, 2019; Kamath et al., 2019).

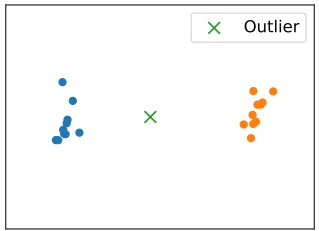 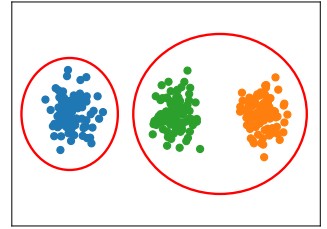 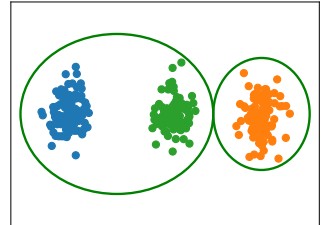

(a) $k$-means Original Dataset      (b) Add data, $c_2$ outlier is OUT      (c) Add data, $c_2$ outlier is IN

Figure 4: The non-convexity of the $k$-means objective can prevent forgetting. When an outlier (green $\times$ in Subfigure 4a) is present in the initial training set, the clustering in Subfigure 4c is learned; when the outlier is not present, the clustering in Subfigure 4b is learned. Adding more data does not result in the outlier's influence being forgotten. Left to right, clusters are $c_1$, $c_2$, and $c_3$ in the text. We add a second dimension to the setting we discuss in the text for presentation.

concrete description of the dataset we consider in Appendix D.1, but we verify empirically that this setting results in near-perfect MI on the outlier point, with 97% accuracy and perfect precision. **This is the first example of privacy leakage we are aware of that relies on non-convexity.** There is a large body of differentially private algorithms which are specific to convex models, and do not apply to non-convex settings (Chaudhuri et al., 2011; Talwar et al., 2015; Wu et al., 2017; Iyengar et al., 2019). Our analysis here is the first to concretely justify this separation.

## 5.2 DETERMINISTIC SGD DOES NOT FORGET

Having observed a (constructed) setting in which forgetting does not happen, we now turn our attention to why forgetting happens when training a neural network. Earlier, we hypothesized that nondeterminism leads to the forgetting we observed empirically. Here, we support this by showing that some unknown nondeterminism is required for forgetting, before continuing in Section 5.3 to show a source of nondeterminism that can cause forgetting.

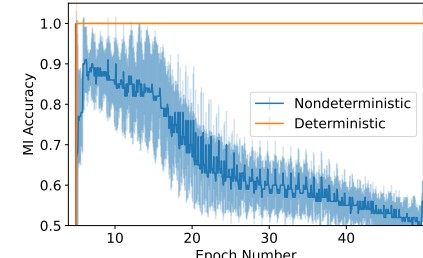

The key idea of this section is that an adversary with knowledge of the entire training set and batch order can simulate the training on $D$ and on $D \cup D_p$, stepping through training in the exact same way as the learner, and receiving the exact same model at each step. Because of the update on $D_p$, these models are never exactly equal, and, because training is deterministic, it is always possible to distinguish them. In Figure 5, we see that this attack is possible in practice, on a logistic regression model trained on the FMNIST dataset with a fixed batch order, using the

Figure 5: For a neural network on FM-NIST, an adversary who knows the exact batch order and training set, doesn't experience forgetting (using INJECT with 1 repeat), with MI accuracy always perfect, even after 45 epochs of updates. Without this perfect knowledge, forgetting happens steadily.

INJECT strategy. By simulating training, the adversary achieves 97% MI accuracy for examples seen in Epoch 5, even after 95 epochs of training (we expect the 3% drop to be a result of unavoidable GPU nondeterminism). In Appendix D.2, we prove that this attack results in perfect MI on a deterministic version of SGD for mean estimation. In this mean estimation setting, adding Gaussian noise to the updates would *provably ensure* forgetting for $D_p$ using the results of Feldman et al. (2018), so it is the nondeterminism itself which prevents forgetting.

## 5.3 RANDOM SAMPLING LEADS TO FORGETTING

Having shown that forgetting doesn't happen when all nondeterminism is known to the adversary, we now show that some amount of unknown nondeterminism can cause forgetting. While Feldman et al. (2018) show that differentially private noise is a sufficient level of nondeterminism to provably cause forgetting, we consider nondeterminism coming from randomized data sampling.

We consider mean estimation, where our examples are drawn from a $d$-dimensional Gaussian distribution with mean $\mu$ and covariance $\Sigma$, i.e., $\mathcal{D} = \mathcal{N}(\mu, \Sigma)$. We are asked to produce a guess for the mean $\tilde{\mu}$. We estimate the mean with gradient descent with learning rate $\eta$, and we ask: does forgetting occur in mean estimation when the data is randomly sampled from this distribution?

We show that it does, by considering a simple distinguishing test. Consider two training runs, both starting SGD from the same fixed initial parameters $\theta_0$. At the very first timestep, we inject two different arbitrary examples, $v$ and $v'$, into the two models. That is, we compute $\theta_0^+$ by taking a gradient step on $v$, and $\theta_0^-$ by taking a gradient step on $v'$. We then continue SGD on randomly sampled examples from $\mathcal{D}$. We write the $k^{\text{th}}$ step of each run as $\theta_k^+$ (for those continuing from $\theta_0^+$) and $\theta_k^-$ (respectively). In Theorem 1, we bound the Rényi divergence between the distributions of $\theta_k^+$ and $\theta_k^-$. Rényi divergence is the divergence used in Rényi DP (Mironov, 2017), and is known to bound the success of MI attacks and reconstruction attacks (Stock et al., 2022), so a smaller Rényi divergence implies that these attacks will perform worse, and the model will forget which of $v$ and $v'$ was used in training. To simplify the analysis, we use $v' = -v$, although a similar result can be shown for arbitrary pairs of $v$ and $v'$.

**Theorem 1.** The distributions of the models $\theta_k^+$ and $\theta_k^-$ produced with learning rate $0 < \eta < 1/2$ have Rényi divergence of order $\alpha$ at most $\frac{2\alpha}{k} v^\top \Sigma^{-1} v$.

Theorem 1 states that distinguishing between the models $\theta_k^+$ and $\theta_k^-$ becomes harder as training progresses, as well as when $v^\top \Sigma^{-1} v$ is small (when $v$ is more "in-distribution"). These agree with observations made in our experiments: forgetting improves as $k$ increases (although with diminishing returns), and more worst-case (or heavily repeated) examples are forgotten more slowly (here, if $v^\top \Sigma^{-1} v = 0$, forgetting never happens!). We note that this theorem is not operational, as it is impossible to know whether the Gaussian assumption holds in practice (Steinke & Ullman), but our analysis provides intuition for our experiments, suggesting further that this source of nondeterminism may lead to the empirical forgetting we observe. Combined with Theorem 2 in Appendix D.2, which also considers mean estimation, Theorem 1 shows that it is the unknown nondeterminism coming from the data which causes forgetting.

## 6 CONCLUSION

In forgetting, we demonstrate a passive form of privacy amplification: examples used early in model training may be more robust to privacy attacks. Our findings align with the theoretical findings of Feldman et al. (2018) (for convex optimization); improved bounds on privacy leakage for early examples also translate to better empirical privacy. Our analytical and empirical findings highlight some previously poorly understood factors underlying forgetting. The size of the dataset is key: forgetting is more likely to happen when (a) the learning algorithm randomly samples training examples at each step (e.g. as in SGD) and (b) these examples are sampled from a large training set (e.g., when training modern language models). Training sets should be large enough for an example to not be seen for thousands of steps before releasing the model. These findings will be useful to practitioners auditing privacy provided by their training algorithm and complement formal analysis of their guarantees (e.g., using DP). Our work suggests the following directions for future investigation.

**Protecting the final examples.** Our work suggests that, for large scale training algorithms, the most recently seen examples are most vulnerable to privacy attacks. Defenses may take advantage of this to improve empirical privacy by focusing their privacy efforts on the most recently seen examples. The number of examples to protect could be determined with an empirical forgetting analysis.

**Non-convexity-aware MI.** In route to understanding forgetting, we demonstrated the first setting where the optimum that a model ends up at leaks information about training examples. While this attack only applies to a specific $k$-means setting, it is interesting whether non-convexity can be exploited to design new MI attacks on realistic tasks.

**Limitations.** While our empirical investigation uses state-of-the-art attacks, it remains inherently heuristic, and cannot prove that forgetting happened. However, we have no reason to expect that the trends we identify do not hold against even stronger attacks. Indeed, in some cases our attacks exploit more knowledge than a realistic adversary can access in practice. Further developments in the techniques we rely on here might be taken advantage of to more faithfully measure forgetting.

**Acknowledgements**

The authors would like to thank Rajiv Mathews, Hakim Sidahmed, and Andreas Terzis for helpful discussion.

Katherine Lee's research is supported in part by a Cornell University Fellowship.

Daphne Ippolito's research is supported in part by the DARPA KAIROS Program (contract FA8750-19-2-1004).

The views and conclusions contained herein are those of the authors and should not be interpreted as necessarily representing the official policies, either expressed or implied, of DARPA or the U.S. Government.

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

## A    ALGORITHMS FOR MEASURING FORGETTING

We present the algorithm for POISON in Algorithm 1 and the algorithm for INJECT in Algorithm 2.

---
**Algorithm 1:** Monitoring Forgetting Throughout Training with POISON

---
**Data:** Training Algorithm TRAIN, Clean Dataset $D$, Poison Samples $D_p$, Removal Step $T_I$,
      Initial Model Parameters $\theta_0$, Total Training Steps $T$, Privacy Attack $\mathcal{A}$

**Function** MEASUREFORGETPOISON():
    $\theta_{T_I}^0 = $ TRAIN$(\theta_0, D, T_I)$
    $\theta_{T_I}^1 = $ POISONTRAIN$(\theta_0, D, D_p, T_I)$
    **For** $i = (T_I + 1)..T$
        $\theta_i^0 = $ TRAIN$(\theta_{i-1}^0, D, 1)$
        $\theta_i^1 = $ TRAIN$(\theta_{i-1}^1, D, 1)$
        ACC $= \mathcal{A}(\theta_i^0, \theta_i^1, D_p)$         ▷ Continuously monitor privacy attack success on $D_p$

**Function** POISONTRAIN$(\theta, D, D_p, T_I)$:
    **Return** TRAIN$(\theta, D \cup D_p, T_I)$     ▷ Train with poisoned data over multiple epochs

---

## B    DETAILED EXPERIMENT SETUP

**ImageNet.** We train ResNet-50 models for 90 epochs, with a learning rate of 0.1, momentum of 0.9, and batch size of 256 (making each epoch roughly 5,000 steps). To test privacy, we focus mainly

---

**Algorithm 2:** Monitoring Forgetting Throughout Training with INJECT

---

**Data:** Training Algorithm TRAIN, Clean Dataset $D$, Poison Samples $D_p$, Injection Step $T_I$,
Injection Count $k$, Initial Model Parameters $\theta_0$, Total Training Steps $T$, Privacy Attack $\mathcal{A}$

**Function** MEASUREFORGETINJECT():
    $\theta_{T_I}^0 = \text{TRAIN}(\theta_0, D, T_I)$
    $\theta_{T_I}^1 = \text{INJECTTRAIN}(\theta_0, D, D_p, T_I, k)$
    **For** $i = (T_I + 1)..T$
        $\theta_i^0 = \text{TRAIN}(\theta_{i-1}^0, D, 1)$
        $\theta_i^1 = \text{TRAIN}(\theta_{i-1}^1, D, 1)$
        $\text{ACC} = \mathcal{A}(\theta_i^0, \theta_i^1, X_I, Y_I)$       ▷ Continuously monitor privacy attack success

**Function** INJECTTRAIN $(\theta, D, D_p, T_I, k)$:
    $\theta_{T_I} = \text{TRAIN}(\theta, D, T_I)$
    **Return** $\text{TRAIN}(\theta_{T_I}, D_p, k)$       ▷ Inject after training on clean data

---

on the INJECT strategy, due to the large number of steps made in a single ImageNet epoch. We also briefly compare with POISON. In our experiments, we start from a fixed checkpoint (with no injected data), and fine-tune from this checkpoint for both the injected and baseline models, and perform membership inference attacks to distinguish the injected predictions from the baseline predictions. We calibrate the logit scores based on those produced by the fixed checkpoint, representing a powerful adversary who knows even an intermediate update before the inclusion of sensitive data. We also experiment with calibrated loss scores, and with not performing calibration, and find that these variants of the attack perform worse. We inject a single batch of data, and measure the precision at a false positive rate of 10%, averaging over three trials. We note that, because both the injected and baseline models are initialized from the same checkpoint, the first batch of the injected model will be the injected data, at which point membership inference will reach 100% accuracy/precision for any parameter setting.

**LibriSpeech.** We use the state-of-the-art Conformer (L) architecture and the training method from Gulati et al. (2020) to train models over LibriSpeech. We train each model with a batch size of 2,048 utterances for 100,000 steps using the Adam optimizer (Kingma & Ba, 2015) and a transformer learning rate schedule (Vaswani et al., 2017).We use the POISON strategy, as we make many passes over LibriSpeech. To generate canary utterances, we start with a vocabulary consisting of the top 10,000 words in the LibriSpeech test set. For the canary transcripts, we sample each word randomly from the vocabulary, varying the length of the transcript ($\{7, 10\}$ words), and the number of insertions in the dataset ($\{1, 4, 10, 20\}$). For the canary audios, we use Text-To-Speech generation, and vary the gender of the speaker ($\{$Male, Female$\}$), and the number of speakers per transcript ($\{1, 2\}$ of the same gender). For each canary configuration, we generate 10 unique canaries. As a result, we generate 320 unique canaries that amount to a total of 2,840 utterances ($<$1% of total utterances in LibriSpeech). We also use loss calibration, for which we train 11 reference models using a random 80% partition of LibriSpeech for training each model.

**Large Language Models.** We train decoder-only, 110M parameter, language models (with the T5 codebase and architecture from Raffel et al. (2020)) for one epoch over a version of C4 (Dodge et al. (2021)). In total, this was 83,216 steps with a batch size of 4,096. We deduplicated C4 with the MinHash strategy introduced by Broder (1997) and the implementation and parameters from Lee et al. (2021). Lee et al. (2021) showed that duplicating examples contributes to memorization, thus we seek to start our experiments with a deduplicated dataset. We used a threshold of 0.7 for both edit-distance and Jaccard similarity. We use the INJECT strategy due to the large training set. For each batch of canaries, we created 256 unique canaries consisting of random tokens and duplicated them between 1 and 100 times to generate a full batch of 4,096 examples (including duplicates). Batches of canaries were inserted 10%, 50%, and 90% of the way into one epoch of training. Additionally, we another model, holding random seed, architecture, and data order fixed, without any canaries as a baseline.

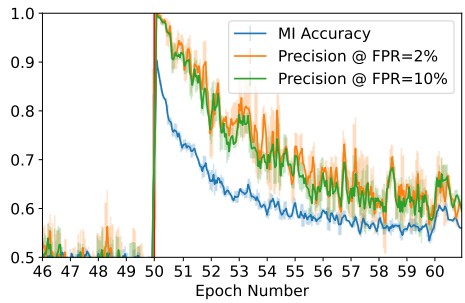

Figure 6: ImageNet figures have some smoothing applied. We present here an unsmoothed version of Figure 1a.

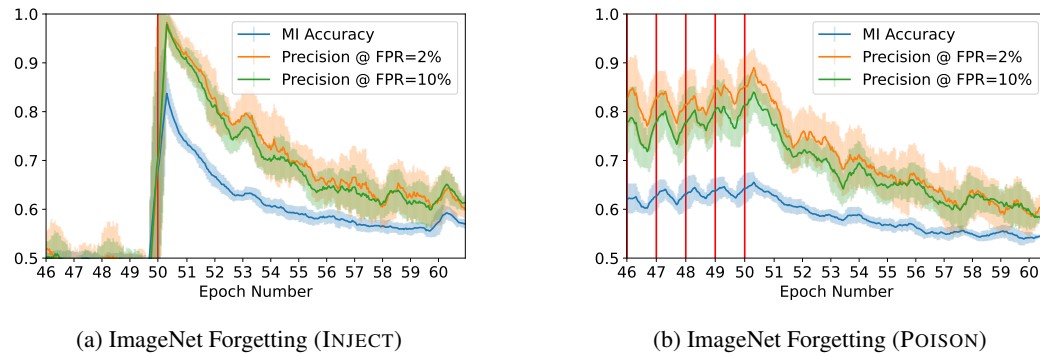

(a) ImageNet Forgetting (INJECT)                              (b) ImageNet Forgetting (POISON)

Figure 7: On ImageNet, the INJECT and POISON strategy result in similar forgetting curves, when the example is used 10 times (all at once in INJECT, and once per epoch in POISON).

## C    EXTENDED EXPERIMENTS

### C.1    POISON ON IMAGENET

We reproduce Figure 1a in Figure 7a, which shows forgetting measured with INJECT on ImageNet with 10 duplicates, and compare it side-by-side with POISON in Figure 7b, where each example is added once per ImageNet epoch, for 10 epochs. We see forgetting happens here at roughly the same rate, after controlling for how many times an example is seen.

### C.2    RECENCY

We plot the impact of recency on forgetting in Figure 8. To do so, we inject samples at various points in training, and measure how quickly models return to a small privacy leakage. We investigate this in Figures 8a, 8b, and 8c for the ImageNet, LibriSpeech, and C4 datasets, respectively. On ImageNet, it appears that the later a point is seen, the longer it takes to forget, although this effect seems to level off for ImageNet (from epoch 40 to 80, there is little change). This is also true for LibriSpeech, where the later a point is seen, the more times it was used in training. However, there is no clear trend on the C4 dataset. Attacks which better exploit the optimization trajectory may find earlier examples to be forgotten more slowly, as they may have a longer impact on this trajectory. For now, privacy attacks appear to not be able to take advantage of the trajectory, and the optimization landscape appears to obfuscate rather than reveal memorized examples.

### C.3    MODEL SIZE

We show the relationship between model size and forgetting in Figures 9a and 9b, for hard test examples and medium hardness test examples, respectively. We evaluate forgetting for ResNet-18, ResNet-34, ResNet-50, and ResNet-101 models. There seems to be little difference between models

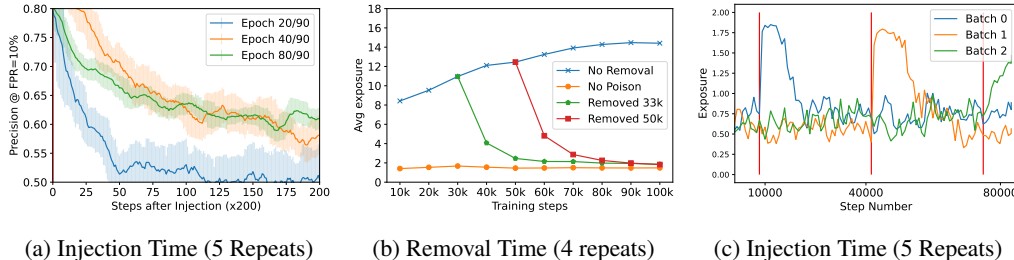

(a) Injection Time (5 Repeats)    (b) Removal Time (4 repeats)    (c) Injection Time (5 Repeats)

Figure 8: There is no consistent effect for injection/removal time. On ImageNet, injection in Epoch 20 leads to earlier forgetting than later epochs. On LibriSpeech, later removal leads to slower forgetting, due to using the POISON strategy. For LMs, there is little difference in exposure between inserting 10% through training and 50%.

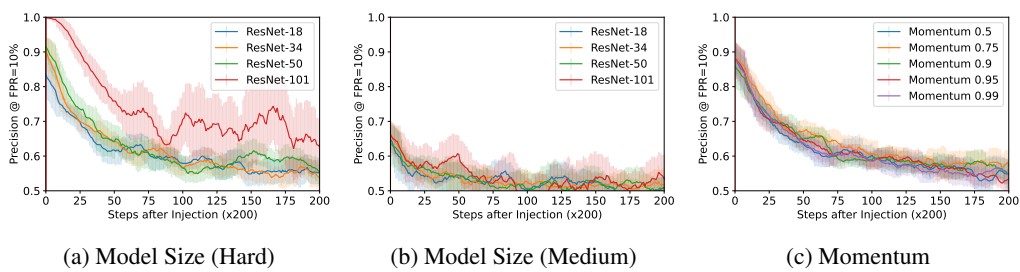

(a) Model Size (Hard)    (b) Model Size (Medium)    (c) Momentum

Figure 9: Hard examples seem to be forgotten more slowly by ResNet-101 models, but medium hardness examples are forgotten quickly by all models. Momentum does not appear to have a significant impact on forgetting. Results on ImageNet.

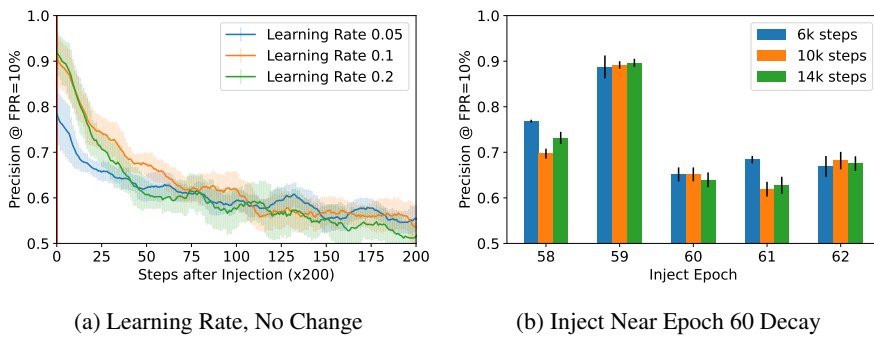

(a) Learning Rate, No Change    (b) Inject Near Epoch 60 Decay

Figure 10: Changing the global learning rate does not have a significant impact on forgetting (left), except immediately before or after large learning rate changes (right). In this model, the learning rate is decayed at Epoch 60. Forgetting happens less quickly at Epoch 59, right before the decay (Epoch 59 injection still has high precision after 14k steps); forgetting happens much more quickly for data injected in Epoch 60 immediately after the learning rate decay.

for medium hardness test examples, but ResNet-101 models seem to forget hard test examples more slowly. Prior work has found that larger models generally memorize more (Carlini et al., 2022), making this effect natural, but it is interesting that there is little difference between most of these models. It is possible that all ResNet models we consider are already highly over-parameterized, and forgetting has already nearly saturated.

## C.4 OPTIMIZER PARAMETERS

On ImageNet, we vary the momentum and learning rate parameters to see if these result in a significant change to forgetting. For example, it is possible that an increased momentum parameter would lead to less forgetting, as it causes a point's update to contribute to every subsequent update. However, we find this to generally not be the case in our experiments on ImageNet: learning rate and momentum do not appear to have a significant impact on forgetting (Figures 9c and 10a, respectively). Learning rate likely does not have a large impact, as all examples have the same learning rate, and so the data nondeterminism is equally powerful at causing forgetting in each case. While it is intuitive that larger momentum parameters would lead to longer forgetting, forgetting happens over long enough step counts for momentum to not impact it significantly. However there is one exception: in our ImageNet training, we use a learning rate decay which reduces learning rate by a factor of 10 every 30 epochs (we reproduce Figure 3 in Figure 10b to discuss this effect more here). This has two drastic implications on forgetting: examples seen just before the decay are forgotten much more slowly, and there is a period directly after the decay where forgetting happens more quickly. The first phenomenon (seen in Epoch 58 and Epoch 59 in Figure 10b) is intuitive: the rate of forgetting depends on the other examples' contributions, and the point injected before the learning rate decay contributes 10x more than these later examples. The other phenomenon is more surprising, and it is likely due to the optimum being more "unstable" after a learning rate decay. This instability reduces over time, as we see in Figure 10b (at Epochs 60-62); forgetting has begun to increase to a normal rate two epochs after the learning rate decay, at Epoch 62.

## C.5 COMPARISON WITH TRADITIONAL FORGETTING METRICS

We have argued that our approach is more relevant to privacy than traditional metrics for measuring forgetting, such as loss or accuracy (accuracy is standard in the catastrophic forgetting literature and was also used to measure forgetting in Tirumala et al. (2022)). Here, we present additional experimental justification. First, we see in Figure 11a that accuracy is an overly optimistic metric from a privacy perspective. For examples repeated 10 times, accuracy has stabilized by $100 \times 200 = 20000$ steps after injection, suggesting that they are completely forgotten. However, compared to Figure 2a, these examples are still at 70% membership inference precision, not invulnerable from privacy attacks. Loss (in Figure 11b) takes longer to stabilize, providing more comparable forgetting speed to our approach. However, membership inference scores are still more easily interpreted than loss scores for quantifying privacy risk. We also note that, to make comparison easier, we have used *hard* examples to report results with repetitions, but existing work typically considers more *average* examples, as forgetting is traditionally used to measure average performance on subdistributions. For this reason, existing metrics are more similar to our experiments with *medium* hardness or *easy* examples in Figure 11c, where forgetting happens nearly immediately, while hard examples take significantly longer. Worst-case analysis is more relevant from a privacy perspective, justifying our approach.

## D EXTENSION OF UNDERSTANDING FORGETTING, SECTION 5

### D.1 CONCRETE k-MEANS DESCRIPTION

We reproduce here Figure 12, which offers intuition for our setting. Concretely, we consider a one-dimensional $k$-means task with $k = 2$, but where three clusters exist in the data, denoted by $c_1, c_2, c_3$. Samples from cluster $c_1$ are drawn from $\mathcal{N}(-1, \sigma^2)$, samples from cluster $c_2$ are drawn from $\mathcal{N}(\mu, \sigma^2)$, and samples from cluster $c_3$ are drawn from $\mathcal{N}(1, \sigma^2)$. Here, $\sigma^2$ is some fixed variance parameter (we use $\sigma = 0.03$), and $\mu > 0$ controls the distance between $c_2$ and the other two clusters, and crucially makes $c_2$ slightly closer to $c_3$ than to $c_1$ (we use $\mu = 0.03$). We consider a two-stage learning procedure, where some dataset $D_0$ is used to learn the clusters initially, and the model is then updated with some new dataset $D_1$. We fix $D_1$ to contain an equal number $n$ of

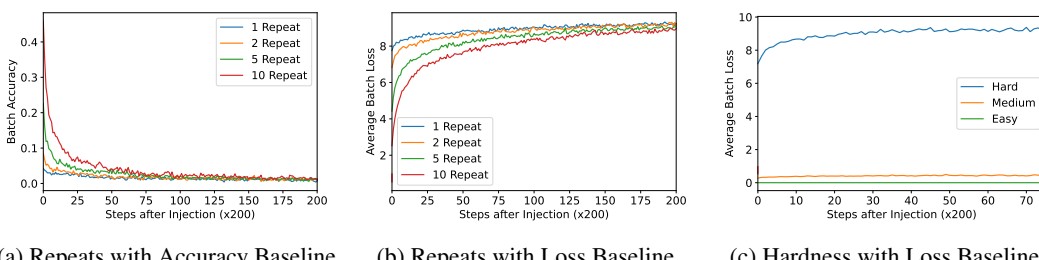

(a) Repeats with Accuracy Baseline    (b) Repeats with Loss Baseline    (c) Hardness with Loss Baseline

Figure 11: Experiments on ImageNet with traditional forgetting metrics - accuracy and loss on the injected batch.

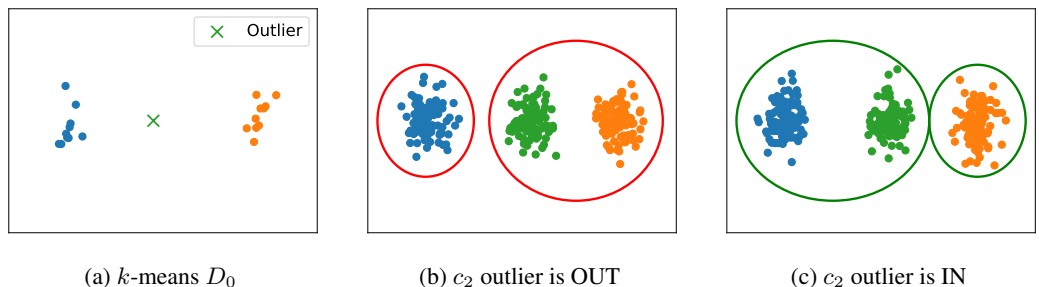

(a) $k$-means $D_0$    (b) $c_2$ outlier is OUT    (c) $c_2$ outlier is IN

Figure 12: The non-convexity of the $k$-means objective can prevent forgetting. When an outlier (green $\times$ in Subfigure 12a) is present in the initial training set, the clustering in Subfigure 12c is learned; when the outlier is not present, the clustering in Subfigure 12b is learned. Adding more data does not result in the outlier's influence being forgotten. Left to right, clusters are $c_1$, $c_2$, and $c_3$ in the text. We add a second dimension to the setting we discuss in the text for presentation.

samples from each cluster. However, we consider two cases for $D_0$: (1) in the IN case, $D_0$ consists of $m$ samples from $c_1$, one "outlier" sample $x$ (we use $x = -0.01$, slightly more than one standard deviation from the mean) from $c_2$, and $m$ samples from $c_3$, and in the OUT case, $D_0$ consists of $m$ samples from $c_1$ and $m$ samples from $c_3$, without the "outlier". Samples from $c_2$ are never included in $D_0$, perhaps because they are from an emergent subpopulation. The single "outlier" sample can result in drastically different clusters, a difference which does not get forgotten by collecting more examples $D_1$. We run an experiment with this setup, using $m = 10$, $n = 100$, and run 200 trials with fresh samples from each cluster. Over these trials, membership inference accuracy for $x$ has 97% accuracy, but the attack also has perfect precision. That is, when $x$ is OUT, $c_1$ and $c_2$ are far enough that they are never joined, and there are only examples where the randomness of the data causes $c_2$ and $c_3$ to join despite the presence of $x$. We note that differential privacy would prevent this attack, as it would randomize the clustering, preventing membership inference. Machine unlearning algorithms designed for $k$-means (Ginart et al., 2019) would also retrain the entire model upon removing the outlier.

### D.2   NONDETERMINISM IS REQUIRED

The analysis in the remainder of this section will use the mean estimation task, a simple task which is frequently used in the theoretical privacy literature to understand the effectiveness of privacy attacks (Homer et al., 2008; Sankararaman et al., 2009; Dwork et al., 2015) or design differentially private algorithms (Bun & Steinke, 2019; Kamath et al., 2019). In mean estimation, a learner wants to find the mean of a dataset, which is the minimizer of the squared $\ell_2$ loss. To understand forgetting, we consider algorithms which perform mean estimation by per-example gradient descent, iterating through samples $x$ from the dataset and updating the model at iteration $i$, $\theta_i$, by a gradient step on the squared $\ell_2$ loss $(\theta_i - x)^2$, with learning rate $\eta$.

To show that determinism prevents forgetting, we consider the deterministic version of per-example gradient descent in Algorithm 3, which iterates through a dataset in a fixed order. In Theorem 2, we

prove that this algorithm doesn't forget, by considering two datasets which differ in only one row, and showing that the two datasets have perpetually different models, so that an adversary can distinguish between them if the adversary knows all other points. In the main body, we also experimentally verify this result, when logistic regression is trained by minimizing the standard cross-entropy loss with fixed-order SGD on the FMNIST dataset.

---

**Algorithm 3:** Mean Estimation with Deterministic Ordering

---

**Data:** Dataset $D = \{x_i\}_{i=1}^n$, initial parameters $\theta_0$, learning rate $\eta$

**Function** TRAINMEANDETERMINISTIC $(D, \theta_0, \eta)$:

    **For** $i = 1..n$

        $\theta_i = \theta_{i-1} - 2\eta(\theta_{i-1} - x_i)$

    **Return** $\theta_n$

---

**Theorem 2.** Consider the described algorithm $T$ in Algorithm 3, with $0 < \eta < 0.5$. When run on any two distinct datasets $D_0$ and $D_1$ differing in only one row, we have $T(D_0) \neq T(D_1)$, regardless of which index $D_0$ and $D_1$ differ in.

*Proof.* Consider the intermediate points $\theta_i^0$ and $\theta_i^1$, which represent the model at the $i$th step of training on datasets $D_0$ and $D_1$, respectively. Write $j$ for the index where $D_0$ and $D_1$ differ. To prove the theorem, we first note that $\theta_{j-1}^0 = \theta_{j-1}^1$, as the first $j-1$ steps of training all use identical examples from $D_0, D_1$. Next, we show that $\theta_j^0 \neq \theta_j^1$, and then that $\theta_i^0 \neq \theta_i^1$ for all $i > j$.

To see that $\theta_j^0 \neq \theta_j^1$, we write out the precise gradient updates:

$$\theta_j^0 - \theta_j^1 = \theta_{j-1}^0 - 2\eta(\theta_{j-1}^0 - x_j^0) - (\theta_{j-1}^1 - 2\eta(\theta_{j-1}^1 - x_j^1)) = 2\eta(x_j^0 - x_j^1) \neq 0,$$

where the second step holds because $\theta_{j-1}^0 = \theta_{j-1}^1$, and the third holds because $j$ is the index where $D_0$ and $D_1$ differ.

Now, to show that this implies that $\theta_i^0 \neq \theta_i^1$ for all $i > j$, we show that, for all $x$, the gradient update function is one-to-one. That is, for all $x$, $\theta \neq \theta'$, taking a gradient step on $x$ still results in different models:

$$\theta - 2\eta(\theta - x) = \theta(1 - 2\eta) + 2\eta x \neq \theta'(1 - 2\eta) + 2\eta x = \theta' - 2\eta(\theta' - x).$$

From this, $\theta_j^0 \neq \theta_j^1$ implies that the models are different at step $j + 1$, which itself implies that the models are different at step $j + 2$, and will be different for the entirety of training. $\square$

### D.3 DATA RANDOMNESS CAN CAUSE FORGETTING

Here, we prove Theorem 1, restated below.

---

**Algorithm 4:** Mean Estimation with Random Sampling and Injection

---

**Data:** Injection $v$, initial parameters $\theta_0$, training steps $k$, distribution $\mathcal{D} = \mathcal{N}(\mu, \Sigma)$, learning rate
    $\eta$

**Function** TRAINMEANSAMPLED $(\theta_0, v, k, \eta, \mathcal{D})$:

    $\theta_0^+ = \theta_0 - 2\eta(\theta_0 - v)$             $\triangleright$ Update with $+v$

    $\theta_0^- = \theta_0 - 2\eta(\theta_0 + v)$             $\triangleright$ Update with $-v$

    **For** $i = 1, \ldots, k$

        $x_i^+, x_i^- \leftarrow$ SAMPLE$(\mathcal{D})$

        $\theta_i^+ = \theta_{i-1}^+ - 2\eta(\theta_{i-1}^+ - x_i^+)$

        $\theta_i^- = \theta_{i-1}^- - 2\eta(\theta_{i-1}^- - x_i^-)$

    **Return** $\theta_k^+, \theta_k^-$

---

**Theorem 1.** *The distributions of the models $\theta_k^+$ and $\theta_k^-$ output by Algorithm 4 with learning rate $0 < \eta < 1/2$ have Rényi divergence of order $\alpha$ at most $\frac{2\alpha}{k} v^\top \Sigma^{-1} v$.*

*Proof.* First, observe that the gradient update on some $\theta_0$ with example $x_0$ produces $\theta_1 = \theta_0 - 2\eta(\theta_0 - x_0) = \theta_0(1 - 2\eta) + 2\eta x_0$. Another gradient update on $x_1$ produces $\theta_2 = \theta_0(1 - 2\eta)^2 + 2\eta(1 - 2\eta)x_0 + 2\eta x_1$. In general, we have

$$\theta_k = \theta_0(1 - 2\eta)^k + 2\eta x_{k-1} + 2\eta(1 - 2\eta)x_{k-2} + \cdots + 2\eta(1 - 2\eta)^{k-1}x_0. \tag{1}$$

We write $\theta_0^+ = \theta_0(1 - 2\eta) + 2\eta v$ (a gradient step on $+v$) and $\theta_0^- = \theta_0(1 - 2\eta) - 2\eta v$ (a gradient step on $-v$). Then we analyze the distributions $\theta_k^+$ and $\theta_k^-$. Using Equation 1, replacing $\theta_0$ with $\theta_0^+$ and $\theta_0^-$, and using that each $x_i$ is sampled from $\mathcal{N}(\mu, \Sigma)$, we obtain the distributions

$$\theta_k^+ = \mu + (\theta_0^+ - \mu)(1 - 2\eta)^k + \mathcal{N}\left(0, \frac{\eta(1 - (1 - 2\eta)^{2k})}{1 - \eta}\Sigma\right),$$

$$\theta_k^- = \mu + (\theta_0^- - \mu)(1 - 2\eta)^k + \mathcal{N}\left(0, \frac{\eta(1 - (1 - 2\eta)^{2k})}{1 - \eta}\Sigma\right).$$

Using that the Rényi divergence of order $\alpha$ between two multivariate Gaussians with means $\mu_0, \mu_1$ and equal covariance $\Sigma'$ is $\frac{\alpha}{2}(\mu_0 - \mu_1)^T\Sigma'^{-1}(\mu_0 - \mu_1)$, we can compute the divergence for $\theta_k^+$ and $\theta_k^-$ as $8\alpha\frac{\eta(1-\eta)(1-2\eta)^{2k}}{1-(1-2\eta)^{2k}}v^T\Sigma^{-1}v$.

It remains to bound the $\eta$ term in this product. The function $f(\eta) = \frac{\eta(1-\eta)(1-2\eta)^{2k}}{1-(1-2\eta)^{2k}}$ has negative derivative on the interval $0 < \eta < 1/2$ for any $k > 0$. Then $f(\eta)$ it is maximized on $0 < \eta < 1/2$ when $\eta \to 0$; we can compute this limit with L'Hospital's rule as $\frac{1}{4k}$, completing the proof. $\qquad\square$

