# OpenReview forum: "Measuring Forgetting of Memorized Training Examples"
_ICLR.cc/2023/Conference — ICLR 2023 poster_

### Official Review · Reviewer_2BVh · 2022-10-24

**Confidence:** 3
**Correctness:** 3
**Technical Novelty And Significance:** 3
**Empirical Novelty And Significance:** 3
**Recommendation:** 6

**Clarity, Quality, Novelty And Reproducibility:**


Clarity: The paper is overall well-written but there are still some issues (see the above points).

Quality: Overall, the paper is of good quality (except for the above points).

Reproducibility: the code is not provided.

Novelty: Some insights illustrated in the experimental results are already known in the prior work, but the systematic experiments conducted in this paper are still useful.


**Strength And Weaknesses:**


* The hypotheses on why forgetting happens in practice is interesting and could spur future research on this topic

Weaknesses:

* The empirical results on forgetting are not surprising. In fact, as discussed in Section 2.3, similar insights have already been discussed in prior work. (Although there is still value in demonstrating the empirical results).

* Overall the paper is well-written. However, there are still some unclear points:
    - Page 4: "we instead design a procedure to measure an algorithm’s worst-case forgetting." How can this algorithm measure the *worst-case* forgetting?
    - Figure 1a: the accuracy increases before the injection (the vertical line). How could that happen?
    - Page 6: "...by differential privacy with ε ≈ 0.6". How do you get this epsilon? The detail is missing.
    - The last paragraph of Section 4: "We ﬁnd that examples injected before the decay are forgotten signiﬁcantly slower than examples injected after." I do not see how to draw this conclusion from Figure 3.
    - Continuing the above question: figure 3 uses #steps to label the bars, but uses inject epoch as the x-axis. Without knowing the number of steps per epoch, it is unclear how to read this figure.
    - The discussion in Section 5.1 is interesting. However, it slightly disconnects from the main story of the paper. While the non-convexity in the illustrated k-means example "can prevent forgetting", it does not give much insight into the neural networks, as the non-convexity in the two settings is very different. The story goes much better without Section 5.1.
    - Section 6: "We ﬁnd the size of the dataset is key". Which results in the main text illustrate this?
    - Section 6: "forgetting is more likely to happen when ... these examples are sampled from a large training set (e.g., when training modern language models)." Which results in the main text illustrate this?


**Summary Of The Paper:**

The paper empirically evaluates how machine learning models forget the training samples during the training process. The paper empirically demonstrates that machine learning models do memorize the training samples once it sees them, but tend to forget them gradually once the training proceeds without seeing them again. The paper claims that the non-determinism in the training process might be one possible reason, by showing that empirically and theoretically that forgetting does not happen in deterministic training, but could happen when non-determinism is introduced.

**Summary Of The Review:**


Although the insights illustrated in the experiments are not surprising, having such a systematic empirical study is still useful to the community. The proposed metric can be useful for follow-up work. The hypothesis on the reason for forgetting can spur future research.

---

> ### Author Response · Authors · 2022-11-11
> **Thank you for the review!**
>
> Thank you for your time in reviewing our work! We’ve added a revised version of the paper, with changes in blue.
>
> Unsurprising: It indeed appears intuitive that examples seen more recently would enjoy less privacy. This is not a novel argument that our paper makes. Rather, this is the *hypothesis* that we set out to rigorously test and validate empirically in this work. Our novel contribution is to quantitatively measure the strength of this forgetting effect in various settings. We hope our results will open up new avenues for decreasing the utility cost of learning with privacy guarantees.
>
> “Worst-case forgetting”: More accurately, we attempt to measure worst-case forgetting, using strong privacy attacks. We cannot reach any provably worst case analysis, as is the nature of empirical privacy work. We’ve reworded this sentence in Section 3.2.
>
> Accuracy increases before injection: This is a plotting artifact - we had some smoothing enabled to make it easier to read. We’ve added an unsmoothed plot in the new Figure 6 in Appendix C.
>
> DP Epsilon: We computed this simply as ln(Precision/(1-Precision)) (with 50/50 IN/OUT split, this is identical to TPR/FPR, which is the typical way of converting between DP epsilon and attack success rates, see Kairouz et al. 2015, Jagielski et al. 2020, Nasr et al. 2021). We did not use any statistical confidence (as is done in work on auditing DP guarantees) as this number was simply meant to contextualize a given attack success rate. Do you think this should be phrased differently?
>
> Interpreting Figure 3: The bar chart shows the precision after a certain number of steps (at 0 steps, precision is 100%). At Epoch 59, after 14k steps, precision is still ~90%. Forgetting is very slow. By contrast, for injection at Epoch 60 (directly after the learning rate decay), only 6k steps are required for forgetting to result in <70% precision.
>
> Steps per epoch: An epoch is roughly 5000 steps.
>
> Nonconvexity is different in kmeans and NN: There is a difference between kmeans and neural networks, but our kmeans result is a first step, showing that nonconvexity-specific privacy attacks exist. We leave an extension to neural networks as a (very!) interesting question for future work.
>
> Conclusion “size of the dataset” mentions: The comment here is implicit throughout the paper - the dataset must be large enough for O(10k) steps. We’ve added a sentence clarifying this in the conclusion.

---

> > ### Comment · Reviewer_2BVh · 2022-11-21
> > **Thank you!**
> >
> > Thank you for the detailed explanations! This rebuttal clarifies most of my concerns. But I still have a question about DP epsilon.
> >
> > Firstly, if you want to mention the epsilon value, you should consider adding the way you compute it and the citations to that sentence.
> >
> > Secondly, I still do not fully understand why it makes sense here. When the two datasets differ in only one sample and the training algorithms for the two models are the same, I can see the reasoning behind this estimation. But in your experiments, the datasets used to train the injected and baseline models differ by more than one sample. In addition, the training procedures for these two models are different. I don't understand why this estimation makes sense here?
> >
> > It would be great if the authors could clarify.

---

### Official Review · Reviewer_TqY1 · 2022-10-24

**Confidence:** 3
**Correctness:** 3
**Technical Novelty And Significance:** 3
**Empirical Novelty And Significance:** 3
**Recommendation:** 6

**Clarity, Quality, Novelty And Reproducibility:**

The paper is well-written and easy to read. There lacks some empirical evidence to show why the proposed forgetting measure method is better than traditional measuring methods. The appendix includes details on the experiment implementation.

**Strength And Weaknesses:**

Many thanks for the paper. I thoroughly enjoyed reading this paper and thinking about how privacy risk is related to forgetting and memorization problems in deep learning. I listed the strengths and some of my concerns of the paper as follow.

Strength:
+ The paper studies an interesting problem which is the connection between data forgetting and memorization.
+ Different from current forgetting measuring for catastrophic forgetting, this paper proposes a novel method based on privacy attack: an example that is easier to be inferred by a privacy attack is less likely to be forgotten by models. The evaluation results show that, if examples have not been seen recently, they are less susceptible to privacy attacks.
+ This paper also shows that the k-means method is resilient to forgetting and non-deterministic can be the reasons behind forgetting in training.
+ The writing of the paper is clear and good to follow.

Weakness:
+ The paper proposes to use a privacy attack to evaluate the forgetting of examples in training, but I am not quite convinced why the proposed forgetting measure is better than the existing forgetting measuring method. Although section 3 talks about the difference, there is no empirical evaluation to compare this difference. What could be the connection between the privacy-based forgetting metric and the traditional forgetting metric (like accuracy or training loss)? If we use the traditional forgetting metric, can we get similar conclusions discussed in the paper? Like, if examples have not been seen recently, the examples are less likely to be forgotten?

+ I am not sure if the proposed method indeed measures the forgetting of examples. By measuring by inference attack, it is more like measuring how much models memorize examples. Examples never learned by models have low privacy risks but should not be treated as forgotten examples. I feel that the proposed method measures the privacy risk, but not necessarily if models forget examples.

+ Why do we need to “the worst-case” examples to measure forgetting? Is the proposed method a general method to measure forgetting of all training examples (i.e. “average-case” in the paper)?

+ One of the main arguments in this paper is that, if examples have not been seen recently, they are less susceptible to privacy attacks, which may not be so surprising. Privacy leakage on datasets usually comes from unwanted memorization of training data. Examples are expected to be less memorized if they have not been seen recently due to catastrophic forgetting features of deep learning models, which makes them less susceptible.


**Summary Of The Paper:**

This paper studies the connection between training data memorization and data forgetting. By measuring the forgetting with privacy attacks (member inference attack and data extraction attack), this paper shows that examples that are not observed by models in recent training iterations are more likely to be forgotten (i.e. more robust to privacy attack). In addition, this paper also conducts some studies to better understand forgetting and shows that the k-means algorithm does not experience a forgetting and non-deterministic (like random sampling) may be one of the reasons causing forgetting in the training.

**Summary Of The Review:**

I thoroughly enjoyed reading this paper and thinking about how privacy risk is related to forgetting and memorization problems in deep learning. My major concern is that, as a novel method to measure forgetting, there may lack some evidence to show how the proposed method is better than existing methods. Besides, the reason behind the design of the measuring method is a bit unclear.

---

> ### Author Response · Authors · 2022-11-11
> **Thank you for the review!**
>
> Thank you for your time in reviewing our work! We’ve added a revised version of the paper, with changes in blue.
>
> W1 - Comparing with traditional forgetting metrics: We have added a comparison to more traditional metrics of forgetting in Appendix C.5, accuracy (as used by the catastrophic forgetting literature and Tirumala et al.’s measurement of label memorization), and loss (which, as far as we are aware, has not been used to measure any notion of forgetting). We make three remarks. First, from a privacy perspective, existing metrics are less interpretable - changes in loss or accuracy can make it easy to see if something is “forgotten” or “not forgotten”, but it’s difficult to say, from a privacy perspective, that examples are “forgotten enough”. By using well-known and interpretable definitions of privacy risk, our approach helps interpret better the privacy implications of forgetting. Second, accuracy, the most common metric traditionally for investigating forgetting, underestimates privacy risk - in Figure 11a, batch accuracy has roughly stabilized at 10 Repeats by 100x200 steps after injection. However, as seen in Figure 2a, attack precision is still 70% at this point! And third, some trends have qualitatively similar results, such as the importance of repetitions, while other trends have more complicated interpretations, such as hardness of examples. For hardness, hard examples’ losses stabilize more slowly, but (naturally) stabilize to a much higher loss than easy examples. As a result, using the traditional metric, easy examples might be said to not be forgotten at all, while our work suggests they are forgotten, from a privacy perspective, very quickly. Additionally, typical forgetting metrics consider “average-case” accuracy, which would be similar to our “medium” or “easy” experiment, and which would make forgetting seem significantly faster than it does for our worst-case examples.
>
> W2 - Technique doesn’t measure forgetting: The intent of our work is to investigate the privacy implications of forgetting. We hope that our above discussion comparing our work to existing metrics helps clarify this distinction, showing that these metrics are difficult to interpret from a privacy perspective, optimistic, and result in some different interpretations than existing metrics show.
>
> W3 - “Worst-case”?: Our approach can also measure average case forgetting, by changing how our injected set is constructed. We run some experiments on this “hardness” change in Figure 2c. However, it is generally important from a privacy perspective to measure worst-case privacy, to capture the risk to the most vulnerable examples.
>
> W4 - Unsurprising: It indeed appears intuitive that examples seen more recently would enjoy less privacy. This is not a novel argument that our paper makes. Rather, this is the *hypothesis* that we set out to rigorously test and validate empirically in this work. Our novel contribution is to quantitatively measure the strength of this forgetting effect in various settings. We hope our results will open up new avenues for decreasing the utility cost of learning with privacy guarantees.

---

### Official Review · Reviewer_hxrq · 2022-10-25

**Confidence:** 3
**Clarity, Quality, Novelty And Reproducibility:** Good Quality, novelty, and reproducib…
**Correctness:** 2
**Technical Novelty And Significance:** 3
**Empirical Novelty And Significance:** 3
**Recommendation:** 8

**Details Of Ethics Concerns:**

None.

**Strength And Weaknesses:**

Strength:
- Exploring the forgetting phenomenon is a fundamental topic for the machine learning community, thus, the work is interesting and promising.
- This work is inspiring, as connecting privacy and forgetting is novel to me. One interesting direction may be evaluating the effectiveness of DP algorithms using similar connections.
- This work provides a novel measurement for modeling forgetting.
- The paper is well-written and easy to follow.

I do enjoy reading the paper, but several concerns make me give the current rating. I will raise the score if the authors can address my concerns.

Weakness:
- My main concern is about nondeterminism, i.e., Sec. 5. I think the experimental and theoretical analysis seem weak to support the perspective. Specifically,
1) (Sec. 5.1) the authors claim that non-convexity can prevent forgetting, but the designed experiments are not good, where adding more samples when performing the k-means algorithm has little impact on forgetting samples.  I think the reason should be that the employed algorithm does not change the features of samples, thus, always remember the samples, which is different from the scenario of training DNNs, i.e., updating makes features change. Thus, remembering comes from unchanged features rather than nondeterminism.
2) (Sec. 5.2) further experiments conducted to support the perspective are given in Sec. 5.2, which also seems not good enough. Specifically, the adversary holds the same data D, extra data D_p, and even the same batch order. This means that the adversary knows everything. I cannot understand why the adversary is interested in attacking the target because the adversary even knows exactly what the difference is from D_p.
3) (Sec. 5.3) the authors draw a conclusion using the introduced example of mean estimation: introducing two different samples causes the difference between models, but the nondeterminism can cause forgetting, i.e., hard to distinguish the difference. However, the difficulty in distinguishing actually results from more training iterations, k in Theorem 1, rather than the nondeterminism.
I hope the authors could clarify the above questions.


**Summary Of The Paper:**

This work aims to quantify how much models forget the training samples. To do so, the authors connect the forgetting and privacy, then utilize privacy-related algorithms to measure the forgetting. In addition, the authors claim that nondeterminism could be the potential explanation and conduct some experiments and calculations to support it.

**Summary Of The Review:**

Surprisingly novel, but with limited reasonable support.

---

> ### Author Response · Authors · 2022-11-11
> **Thank you for the review!**
>
> Thank you for your time in reviewing our work! We’ve added a revised version of the paper, with changes in blue.
>
> W3 - Nondeterminism isn’t the cause: Section 5.2 and Section 5.3 are meant to be companions to each other. Section 5.2 shows that complete determinism prevents forgetting, and Section 5.3 shows that the addition of nondeterminism results in forgetting. We believe this is more clear when the setting of Section 5.3 is directly compared against the setting in Appendix D.2 (which also considers mean estimation), which we removed from Section 5.2 for space constraints. These two algorithms differ only in the addition of nondeterminism. We have added a sentence to Section 5.3 which clarifies this. Indeed, more training iterations helps because there is more “unknown entropy” which prevents the adversary from performing inference.
>
> W1 - Examples are remembered because features don’t change: We’ve added an experiment addressing this concern: in the deterministic setting, forgetting doesn’t happen even when features change. Our experiment considers a single layer FashionMNIST neural network (we’ve replaced Figure 5), where the features do change, but forgetting doesn’t happen for the deterministic setting, and does in the nondeterministic setting. Here, the crucial reason features not changing does not cause forgetting is because the adversary is able to anticipate how the features will change, whereas, in a nondeterministic environment, this anticipation is impossible due to an inability to predict future updates. This gap is reflected also in the provable forgetting of Theorem 1 (where forgetting happens *despite* the features not changing) and the provable “remembering” in Theorem 2.
>
> W2 - Adversary knows everything in 5.2: The adversary knows everything, except for whether D_p is used in training or not! In this setting, if some amount of Gaussian noise were added to each step of training, the algorithm would be differentially private, and forgetting would provably happen, using the results of Feldman et al. (2018). Without any source of nondeterminism, though, this forgetting doesn’t happen. We have clarified this in the revision in Section 5.2.

---

> > ### Comment · Reviewer_hxrq · 2022-11-21
> > **Thank you for the responses and the corresponding revision**
> >
> > I appreciate the efforts that the authors have made. The clarification makes the paper clear, accordingly, I adjust the score.

---

### Official Review · Reviewer_Qkp9 · 2022-10-25

**Confidence:** 3
**Correctness:** 3
**Technical Novelty And Significance:** 4
**Empirical Novelty And Significance:** 4
**Recommendation:** 6

**Clarity, Quality, Novelty And Reproducibility:**

Claritya and Quality: This paper is well-written and provide sufficient empirical results to support their claim.

Originality: This work is novel from my perspective. Before this work, there is no study to exactly show when the forgetting happens.



**Strength And Weaknesses:**

Strength:

-  This work is well motivated. In this paper, the authors aim to study the connections between memorization and forgetting in the context of data privacy.
- The paper writing is easy to follow. The authors give a clear definition of forgetting before introducing the measure method, which makes it much clear for the reader.
- This work gives several interesting empirical findings and provides in-depth understandings. Specifically, the authors show the two cases for when the forgetting happens, which is useful for the future investigation.

Weakness:

The contribution of this work is limited due to lack of theoretical guarantee. Although this work provides many empirical studies, it is not clear if the conclusions can extend to other settings, like different model architectures. It would be better if the authors can give any theoretical guarantee.

**Summary Of The Paper:**

In this work, the authors study the effect of the forgetting of machine learning models on privacy. To this end, they focus on validating the two hypothesis, including the pessimistic hypothesis and the optimistic hypothesis. Then, they design a method to measure the example forgetting in the training and show several analytic and empirical findings with this method.

**Summary Of The Review:**

In summary, this work is well motivated and easy to follow. Besides, the work provides in-depth understandings with empirical findings, which, I believe, can bring some new insights to inspire future research. However, the contribution of this work is still limited without theoretical results. I am not an expert in this area, so I give a weak accept for this work before the rebuttal.

---

> ### Author Response · Authors · 2022-11-11
> **Thank you for your review!**
>
> Thank you for your time in reviewing our work! We’ve added a revised version of the paper, with changes in blue.
>
> We would like to point out that our paper makes several contributions in addition to our empirical observations that forgetting does happen in deep learning:
> We connect forgetting with memorization.
> We construct an analytical example of when non-convexity can prevent forgetting from happening.
> We analytically demonstrate how deterministic training does not lead to forgetting.
>
> While it is true that our experiments cannot provably demonstrate diminished privacy risk, they do contribute a more complete picture of both forgetting and memorization in deep learning. While prior work (such as Feldman et al., 2018) has studied forgetting theoretically, their analysis is necessarily limited to the convex setting. We are the first to propose a comprehensive characterization of forgetting using attacks as a medium to approximate the worst-case privacy risk. Furthermore, we do think that the breadth of our experiments (the variety of datasets and models we experimented with) helps provide confidence in the generality of our results.

---

### Decision · Program_Chairs · 2023-01-20

**Decision:**

Accept: poster

**Justification For Why Not Higher Score:**

It may be a spotlight presentation if there is still more room.

**Justification For Why Not Lower Score:**

All reviewers agreed that the novelty and significance are above the bar.

**Metareview: Summary, Strengths And Weaknesses:**

The paper studied the inductive bias of deep learning in particular the memorization and forgetting of training data. The paper proposed a technique to measure to what extent models forget the specifics of training data, applied the findings to privacy, and identified some potential explanation. Although the contents are not theoretically guaranteed, the problem of interest is very fundamental and a lot of researchers/practitioners may be interested in this paper. All reviewers agreed that the novelty and significance are above the bar and thus we should accept it for publication.

Two minor comments: (a) A missing related work is "SIGUA: Forgetting may make learning with noisy labels more robust", ICML 2020; (b) When copying the abstract from latex to OpenReview, you should remove the newlines and make your abstract a single paragraph.

**Note From Pc:**

if the above contains the word "oral" or "spotlight" please see: "oral" presentation means -> notable-top-5% and "spotlight" means -> notable-top-25%. As stated in our emails, we are disassociating presentation type from AC recommendations